# Bee Pollen as Functional Food: Insights into Its Composition and Therapeutic Properties

**DOI:** 10.3390/antiox12030557

**Published:** 2023-02-23

**Authors:** Asmae El Ghouizi, Meryem Bakour, Hassan Laaroussi, Driss Ousaaid, Naoual El Menyiy, Christophe Hano, Badiaa Lyoussi

**Affiliations:** 1Laboratory of Natural Substances, Pharmacology, Environment, Modeling, Health, and Quality of Life, Faculty of Sciences Dhar El Mahraz, University Sidi Mohamed Ben Abdellah, Fez 30000, Morocco; 2The Higher Institute of Nursing Professions and Health Techniques, Fez 30000, Morocco; 3Laboratory of Pharmacology, National Agency of Medicinal and Aromatic Plants, Taounate 34025, Morocco; 4Department of Chemical Biology, Eure et Loir Campus, University of Orleans, 28000 Chartres, France

**Keywords:** bee pollen, composition, medicinal properties, functional food

## Abstract

Bee pollen is a hive product made up of flower pollen grains, nectar, and bee salivary secretions that beekeepers can collect without damaging the hive. Bee pollen, also called bee-collected pollen, contains a wide range of nutritious elements, including proteins, carbs, lipids, and dietary fibers, as well as bioactive micronutrients including vitamins, minerals, phenolic, and volatile compounds. Because of this composition of high quality, this product has been gaining prominence as a functional food, and studies have been conducted to show and establish its therapeutic potential for medical and food applications. In this context, this work aimed to provide a meticulous summary of the most relevant data about bee pollen, its composition—especially the phenolic compounds—and its biological and/or therapeutic properties as well as the involved molecular pathways.

## 1. Introduction

The pollen grain is the male flower’s reproductive organ; in other words, it carries the organ that carries the male gametes to their progenitor cells. It is produced and spread by higher plants as part of their reproductive process [1]. Bee pollen is the final result of the agglutination of pollen grains harvested by worker bees, held together by nectar and/or honey, and gland secretions, and collected at the hive entrance [2]. It is among the most important bee products gaining popularity as a functional food due to its high concentration of bioactive compounds known for their benefits for both mental and physical health, such as proteins, dietary fibers, lipids, carbohydrates, and minerals [3]. Furthermore, bee pollen is known as “the perfectly complete food” due to its strong antioxidant potential and the presence of antioxidant compounds such as polyphenols, flavonoids, carotenoids, and vitamins (A, C, and E) giving this product a high antioxidant potential and making it the latest trend in dietary supplements [4]. Because of this high load of natural bioactive molecules, various scientific studies have reported that bee pollen possesses a wide spectrum of biological properties such as antioxidant [5], hypoglycemic [6], anti-inflammatory [7], antibacterial [8], and anticancer [9]. As a result of all of this, the German Federal Ministry of Health has formally acknowledged bee pollen as a drug. [10]. This paper aims to provide a comprehensive overview of bee pollen in terms of bee pollen harvesting, chemical and nutritional content, and, finally, its biological and therapeutic characteristics. Furthermore, we intended to unveil for the first time the proposed mechanism of action and the involved biomolecular pathways of the various bioactive compounds of bee pollen that are responsible for the improvement of oxidative stress and associated health conditions.

## 2. Methodology

The following online databases were employed to collect the literature data for this paper: Science Direct, Google Scholar, Web of Science, Pub-Med, and Scopus, using the keywords: “bee pollen”; “bee-collected pollen”; “chemical composition of bee pollen”; “therapeutic effect of bee pollen”; “functional effect of bee pollen”, “protective effect of bee pollen”, and “nutritional value of bee pollen”. After collecting and reviewing all selected articles, their general ideas were summarized and used in this review.

## 3. Bee Pollen: From Flowers to the Hive

It is commonly known that nectar and pollen grains are the main sources of nutrients required for the survival and health of bee colonies. Nectar is mainly produced by the nectariferous glands of plants and serves as the raw material for honey production [11], whereas pollen grains represent the plant’s male gametophyte. They take the form of fine dust with tiny particles that vary in color based on the floral origin [12]. Flower nectar provides bees with carbon and nitrogen, while pollen grains provide other dietary components such as lipids, proteins, vitamins, and minerals [13,14].

The flower–bee interaction is a mutualistic relationship in which the flowers reproduce sexually and the bees feed on nectar and pollen [14]. In this respect, plants adopt many techniques to be pollinated and reproduce, including the coloration of their petals and the emission of scents known as pheromones [15]. Worker honeybees perform hundreds of flights to blooms to collect the necessary amount of pollen and nectar [16]. The honeybees’ behavior of gathering nectar and pollen is known as “foraging” and it is highly vital and necessary for the survival of bee colonies. Foraging is a behavior that develops in worker bees between the ages of one and two weeks [17]. Young foraging bees conduct many scouting flights to become acquainted, and at the age of 21 days, they will leave their hives to seek and gather nectar, honeydew, pollen, water, and many essential elements, as well as the resin used to maintain the hive’s asepsis [18,19]. Foraging bees use their proboscis to collect nectar and water by pumping and capillarity; the liquids are stored in the foragers’ crops until they are discharged to the other workers in the hive [20], who then use their hind legs coated with short stiff hairs called “scopae” to squeeze the collected pollen grains into pollen balls using their saliva and honey, which they finally place into their pollen baskets [16]. The majority of bees have developed specialized mechanisms for transporting pollen to their nests and have adjusted their grooming behaviors to transfer pollen from their bodies to the hive [21].

When the foraging bees arrive at the hive, they cover the pollen balls with saliva, then compact the alveoli with a layer of pollen balls and honey, and finally cap them with a layer of wax [22].

Lactic fermentation occurs at this stage due to the participation of lactic bacteria strains that proliferate inside the hives [14]. These bacteria are *Pseudomonas*, which consumes oxygen and creates an anaerobic environment; *Lactobacillus*, which converts carbohydrates into lactic acid; and finally, *Saccharomyces*, which ensures the metabolism of the rest of the sugars that exist in the medium. These reaction chains reduce the environment’s pH, prevent pollen germination, and improve bee pollen absorption capacity and nutritional value [23]. When bee-collected pollen is completely fermented, it becomes “bee bread,” which provides additional proteins for bees, particularly during the period of royal jelly production, as well as nutrition for larvae, and future workers who are fed a diet of pollen, honey, and a small quantity of royal jelly [18].

## 4. Chemical Composition

Bee pollen is one of the magical superfoods due to its extremely wide range of nutritional compounds and microelements. However, this composition may be affected by botanical origin, harvesting season, and storage methods (freeze-drying duration). Considering this large variability, its nutritional and chemical composition has been extensively studied, summarized, and standardized. In this section, we assembled the main macro- and micronutrients of bee-collected pollen.

### 4.1. Main Compound

#### 4.1.1. Water

Several studies have succeeded in quantifying the water content in bee pollen samples, the results obtained being largely variable and dependent on storage conditions (fresh or dried bee pollen), botanical, and geographical origins [24].

In fresh bee-collected pollen, water content varies between 20 and 30% [2]. However, this high humidity is considered a favorable environment for bacterial and fungal growth [25,26,27]. As a result, the bee pollen freezing process must begin immediately after harvesting [28]. Other researchers prefer the nitrogen processing of fresh bee pollen to preserve the optimal microbiological and nutritional properties [29]. Meanwhile, the maximum water content allowed in dried bee pollen depends on the country and must not exceed 4% according to the Brazilian legislation, 6% in Poland and Switzerland, 8% in Argentina, and 10% in Bulgaria [2,24,30,31,32,33]. Thus, the moisture content can be used as a bee pollen quality criterion.

#### 4.1.2. Protein Content

The production of protein for human consumption as well as defining the need for protein, evaluating its quality to meet human needs, and managing the consequences of variations in dietary protein intake are considered major public health issues. Consequently, it is necessary to ensure a sufficient daily supply of protein of good biological quality, since the human body is unable to store it. Therefore, bee pollen, which has high protein content, could guarantee an ideal diet in terms of protein requirement. In addition, pollen is the principal source of protein for bees, providing the necessary elements for their longevity, organ development, larva growth, and body size [34,35]. Proteins also provide essential substances for royal jelly production in the hive [36].The protein content is highly variable between plant species and harvesting geographic areas [37] and varies between 10 and 40 % of pollen dry weight [2].

#### 4.1.3. Amino Acids

Amino acids are crucial not only for protein synthesis but also for the biosynthesis of hormones as well as other molecules with a biological role. Nutritionally, two kinds of amino acids are distinguished, namely, the essential amino acids that the human organism cannot synthesize, and non-essential amino acids that our body has all the machinery to synthesize. In the same context, bee-collected pollen is often considered the “most natural perfect food” because it is a great source of all essential amino acids needed in honey bee and human nutrition [38,39]. This content varies strongly from species to species and depends on botanical and geographical origin, climatic conditions, and nutrient availability in the plant [37,40,41]. Therefore, the amino acid amount can be used as an indicator of freshness, storage, and drying process adequacy [3,42].

The total amino acid content in bee pollen has been quantified by many researchers, and it generally ranges between 108.1 and 287.7 mg/g of bee pollen [43]. Concerning the amino acid profile, De-Melo and Almeida-Muradian have reported twenty-five amino acids, eight of which are essential (valine, leucine, isoleucine, lysine, phenylalanine, threonine, histidine, and methionine). Tryptophan is usually undetectable because of the hydrolysis method employed in the determination of the amino acids. However, tryptophan was detected in Chinese, Slovenian, Spanish, and Italian bee pollen using specific high-performance liquid chromatography methods [44,45,46]. The remaining amino acids are non-essential, such as aspartic acid, alanine, glycine, glutamine, arginine, asparagine, glutamic acid, serine, tyrosine, cystine, cysteine, γ-aminobutyric acid (GABA), ornithine, proline, and homoserine [3]. It has been reported that proline is the most abundant amino acid in dried bee pollen from many countries, while glutamic acid is the main amino acid in freshly collected bee pollen [3,44,47].

#### 4.1.4. Carbohydrates

Bee pollen is composed of pollen mixed with nectar, and the bee’s salivary secretions. Carbohydrates are the major fraction of bee pollen (13–55%), they are mainly polysaccharides and cell wall material [2]. Carbohydrates can be affected by botanical and geographical origin, harvesting methods, and conditioning processes such as high temperature when drying fresh bee pollen [48].

Sugar

Sugar content is the most important quality parameter in bee pollen characterization studies and should not be less than 40 % [2]. Nevertheless, sugars are generally neglected or included in the total carbohydrates which also regroups the dietary fiber and starch [49]. The sugar composition of bee pollen has been assessed by many studies either as reducing sugar [50,51,52,53,54] or as individual sugars [41,44,49,55,56] and all showed a predominance of glucose and fructose as monosaccharides which represented the major amount of sugar fraction. Sucrose, maltose, trehalose, turanose, and melezitose have been identified in previous studies [49,57]. The sugar content and profile can be considerably influenced by nectar added by bees during the packaging and storage of bee-collected pollen [50,58]. Floral source, drying process, and extraction methods can also greatly affect the sugar content [48,56]

Mannitol is a polyol previously identified and isolated in high concentrations from bee pollen collected by stingless bee *Melipona subnitida* from Jandaíra, Brazil, *Tetragonula biroi Friese* from the Philippines, and *Trigona* from Malaysia [41,59,60]. According to these authors, the significant amount of mannitol did not depend on the floral origin, and it is supposed that the previous stingless bee species are capable of converting the glucose and fructose mainly found in flowers into mannitol via their salivary enzymes.

On this basis, sugars can be considered as an additional parameter for establishing quality standards for bee pollen.

Dietary fibers

Dietary fibers describe the soluble and insoluble fraction of fibers from plant-based foods, which include hemicellulose, cellulose, lignin, oligosaccharides, pectins, gums, and waxes; these compounds are resistant to digestive enzymes, thus they are neither hydrolyzed nor absorbed in the intestinal tract [61,62]. Many recent studies have supported the crucial physiological role of dietary fibers in the human body; indeed, they are involved in type 2 diabetes management by the selective promotion of certain gut microbiota [63,64]. A high-fiber diet was found to be effective in many conditions such as obesity-related disorders, cardiovascular diseases, constipation, inflammatory bowel diseases, and colon cancers [65,66,67,68,69]. Regarding this, bee pollen can be a good source of dietary fiber, especially crude fibers. Despite the importance of bee pollen in the human diet, few characterization studies have focused on the determination of the dietary fiber content of bee pollen. According to Compos et al., total dietary fiber should range between 0.3 and 20 g/100 g of bee pollen dry weight [2]. Dietary fiber content has been reported by a few studies. For instance, a recent study carried out on Slovenian bee pollen showed a range of 10–21.4 g/100 g dry weight bee pollen with 73–82% of crude fiber [49]. Brazilian researchers reported an average of 3.6 ± 1.4 g/100 g of dry weight bee pollen [54]. Colombian bee pollen has also been characterized, and the results showed an average of 14.5 ± 3.5 g/100 g dry weight [70], while El-Kazafy recorded that different Egyptian bee pollen showed a range of 0.15 ± 0.01 and 1.70 ± 0.02 g/100 g dry weight bee pollen [47]. Dietary fiber content may vary according to the botanical origin and methods used during hydrolysis.

#### 4.1.5. Lipid and Fat Content

Physiologically, the human body uses a variety of biosynthetic pathways to synthesize lipids; however, some important lipids cannot be obtained through biosynthesis and must be obtained from food. Essential fatty acids (especially omega-3 fatty acids) are involved in many biological functions and play an important role in the prevention of inflammatory and cardiovascular diseases and hormone-dependent tumors. [71,72]. Indeed, bee pollen can be a great source of these compounds since they are crucial for royal jelly production [73]. The lipidic fraction is most attractive for bees, and thus plants with high lipid concentration pollen are more frequently visited [37]. According to Campos et al., lipid content ranges between 1–13 g/100 g [2], while De-Melo and Almeida-Muradian, reported that the total lipid fraction can reach 22 g/100 g [3]. In the same context, Thakur et al. and De-Melo et al. have reported a huge variation of lipid content in monofloral bee pollen from different countries; *Brassica napus* bee pollen from Brazil, China, India, and Greece showed a total lipid content of 7.4%, 6.6 %, 12.38%, and 7.76% respectively; Cistus bee pollen from Italy, Spain, and Greece showed a total lipid content of −1.9%, 7.2%, and 3.80% respectively. *Cocos nucifera* bee pollen from India and Brazil showed 10.43% and 4.6–5.1% total lipid content [3,73].

The lipid profile of bee pollen has been barely investigated while most research studies have focused on the protein, carbohydrate, and antioxidant content. According to Ares et al., carotenoids, steroids, and fatty acids are the main constituents of bee pollen’s total lipid fraction [52]. A study conducted by Li et al. on three monofloral bee pollen samples from China suggested the presence of nine lipid classes, including triglycerides and fatty acids [74].

The fatty acid profile of bee pollen varies between saturated fatty acids, which include mainly the myristic, stearic, and palmitic acids, and unsaturated fatty acids, which include the oleic, α-linolenic (omega-3), and linoleic (omega-6) acids. This fraction is the most dominant in bee pollen [4,44,73,75,76,77]. Other lipid classes such as phospholipids, triterpenes (oleanolic and ursolic acids), and plant sterols (β-sitosterol) have been isolated from bee pollen in smaller amounts [4,36,76]. All these studies have reported that the lipid content in the studied bee pollen depends on botanical origin, harvesting season, drying, storage, and beekeeping methods.

### 4.2. Micronutrients

Micronutrients include minerals and vitamins that are not involved in the energetic balance but are essential for all chemical reactions and for the maintenance of life. Micronutrients are required in small amounts by the body for its growth and development from birth to old age. In a recent report (2020), most food and health organizations estimated that more than two billion people globally suffer from micronutrient deficiency occurring due to an insufficient intake or impaired absorption of vitamins and minerals [78,79]. Generally, micronutrient deficiency is considered a global health concern for all ages. During pregnancy, this deficiency has a devastating effect on both the mother and her fetus, being associated with anemia, hypertension, gestational diabetes, thyroid disorders, obstetric complications, and failure in the growth and development of the fetus, among other conditions [80,81,82,83,84]. During childhood, micronutrient insufficiency may affect the mental and physical development of the children and increase their vulnerability to and exacerbation of diseases such as impaired host defense and infections, developmental disabilities, autism, ocular disorders, and general loss of energy and potential [85,86,87]. The elderly population is also vulnerable to micronutrient deficiencies, which can lead to many age-related diseases such as mild cognitive decline, high risk of type 2 diabetes, cardiovascular diseases, acute respiratory infections especially coronavirus infection, and immune function impairments [88,89,90,91]. Based on what has been stated above, it is clear that maintaining a micronutrient-rich diet may provide enough protection against all of these pathologies. This protection can be guaranteed by consuming a large variety of well-balanced and rich natural products such as bee pollen.

#### 4.2.1. Minerals

There are around twenty minerals that are essential in the human diet and are classified as macro-elements and oligo-elements, also known as trace elements. Mineral deficiency in the human body causes several metabolic problems and severe developmental defects in pregnancy, and significantly affects the individual’s wellness and economic output [92]. Bee pollen is a good source of essential minerals for the development of bees as well as humans, which represent 2–6 % of its content, with about 25 elements [2]. This makes bee pollen an interesting value-added product.

Potassium (K) is the principal mineral element found in high concentrations in bee pollen (400–2000 mg/100 g of bee pollen), and 15 g of bee pollen covers up to 25% of the recommended daily intake (RDI) of this element (2000 mg/day). Phosphorus (P) is the second element mainly present in bee pollen (0.80–6 mg/100 g of bee pollen), covering 16% of the RDI (1000 mg/day) of 15 g of bee pollen. The third important element is magnesium (Mg; 20–300 mg/100 g of bee pollen) which covers up to 23% of the RDI of this element (350 mg/day) of 15 g of bee pollen. Calcium (Ca) is also widely present in pollen (20–300 mg/100 g of pollen) and covers 7% of the RDI (1100 mg/day) of calcium. These elements are known for their crucial role in bone tissue formation by maintaining the proper osmotic pressure of blood as well as cellular fluids. Iron (Fe), zinc (Zn), copper (Cu), and manganese (Mn) are also microelements present in large quantities in bee pollen, covering up to 37%, 79%, 36%, and 85% of the respective RDIs. These trace elements play an important role in blood formation and also in the growth, development, and reproduction process [2,28,93,94].

There are other trace elements such as cobalt (Co), selenium (Se), molybdenum (Mo), and boron (B) which have been identified in bee pollen from different countries. Adequate intake of these trace elements is necessary to support bone and brain health, they play a key role in the maintenance of vitamin structure, reproduction, thyroid hormone metabolism, DNA synthesis, and protection against oxidative damage and infections [3,95,96,97].

Sodium (Na) is also a macro-element present in bee pollen; however, its content remains below 2 g/kg with a high K/Na ratio, and this ratio makes bee pollen beneficial and safe for daily diets with a good electrolyte balance [3]. The mineral content of bee pollen is recommended as a distinct marker of its floral and geographical origin as well as its quality [5,73].

#### 4.2.2. Vitamins

Vitamins are a class of nutrients or organic compounds essential for the body not synthesized by humans, except for vitamins D, K, and biotin (B7), where vitamin D is synthesized in the body by irradiating skin sterols with UV rays, while biotin and vitamin K are present in certain foods but can also be synthesized by the human intestinal flora. Therefore, the essential vitamins must be daily ingested from food to prevent metabolic disorders related to vitamin deficiencies due to their major role in the synthesis of vital cofactors, enzymes, and metabolic reactions based on coenzymes [98,99,100]. Bee pollen is considered a “vitamin bomb” due to the presence of almost all vitamins with an average of 0.02–0.7% of its total content, with a higher amount of water-soluble than fat-soluble vitamins [4,101]. Table 1 summarizes the different vitamins identified in different samples of bee pollen with different floral species and geographical origins.

water-soluble vitamins

B vitamin group (thiamin (B1), riboflavin (B2), PP vitamin or niacin (B3), pantothenic acid (B5), pyridoxine (B6), biotin (B7), folic acid (B9), and cobalamin (B12)) are the most commonly identified class of water-soluble vitamins in bee pollen (Table 2) [29,75,104,105,107]. Vitamin C or L-ascorbic acid is marginally identified because of its deterioration by thermal pretreatments [28]. Water-soluble vitamins, such as B-complex, are not normally stored in the body in significant amounts, necessitating a daily intake of these vitamins. This group of vitamins plays a key role in host immunity, dermatology, and cellular energy production (B1, B2); they facilitate the production of amino acids and improve their metabolism (B6); and help the body to convert carbohydrates into glucose (B3 or PP). Water-soluble vitamin deficiencies triggered by malnutrition can be the origin of certain metabolic and nervous pathologies [108]. Regarding its high content of water-soluble vitamins, bee pollen could be one of their potential sources.

Fat-soluble vitamins

Bee pollen contains fat-soluble vitamins such as vitamin D, K, E, and A (β-Carotene) with low and variable amounts depending on the botanical origin and the season of collection [73,109,110].

The family of vitamins E is commonly called tocochromanols (tocopherols and tocotrienols), and pollen contains in particular the group of tocopherols (α tocopherol, β-tocopherol, γ-tocopherol, and δ tocopherol) with a dominance of α and γ-tocopherol [56,111]. Vitamin K2 (Menaquinone-4), and two types of vitamin A (β-carotene and retinol) were also detected by Bayram et al. [102]. Although there are no identification studies of vitamin D listed in the international literature, Campos, Komosinska-Vassev, and Khalifa et al. cited the presence of vitamin D in their review articles [36,110,112]. Fat-soluble vitamins are involved in a multitude of physiological processes such as vision, bone health, immune function, and coagulation [113].

#### 4.2.3. Carotenoids

Carotenoids are a highly diversified group of yellow- to red-colored polyenes responsible for the colors in many plant-derived products and play an important role in human health [94]. In bee pollen, β-carotene is the most frequently identified of this class. It is an antioxidant provitamin with good effects on human health (anti-tumor, anti-leukemic, and beneficial against cardiovascular diseases) [114]. Besides β-carotene, other carotenoids such as lutein and cryptoxanthin, zeaxanthin, β-cryptoxanthin, and α-carotene have been identified [60,115]. The carotenoid content varies according to the botanical origin, harvest period (climate conditions), drying, and storage condition [2,52,107].

### 4.3. Pollen Probiotics

Throughout history, bee pollen has been considered a complete food with many therapeutic virtues, and for this reason, it has been the subject of numerous and diverse biochemical and microbiological studies. Many studies have been concerned with the anti-microbial activities of bee pollen, but little is known about its microbiome.

P. Percie du Sert has reported that bees raise lactic ferments in the nectar stored inside the hive, and this bacteria-rich nectar will be used during their flight to stick pollen grains on their forelegs, which explains the presence of lactic acid bacteria in freshly harvested pollen [29]. Several studies have demonstrated that honey bees and bumblebees seem to have a simple intestinal bacterial fauna which includes acidophilic bacteria, mainly from the Lactobacillus family such as *Lactobacillus kunkeii*, *Lactobacillus plantarum*, *Lactobacillus fermentum*; *lactobacillus kunkeii*; *Lactobacillus plantarum*; *Lactobacillus fermentum*; *Lactococcus slactis*; *Pediococcus acidi lactici*; *Pediococcus pentosaceus*; *Lactobacillus ingluviei*; and *Weissella cibaria* [116,117,118,119]. It is now quite clear that the fresh bee pollen bacterial community comes from the specific bacterial fauna of the bee intestinal gut.

Previous in vitro studies have reported the beneficial effect of the isolated probiotic strains of fresh pollen against pathogenic Gram-positive and Gram-negative bacteria through the high production of bacteriocins, such as organic acids, which can be of major importance in fighting human infectious diseases [116,120,121]. Lactobacillus strains isolated from fresh bee pollen can survive under human digestive tract conditions such as low pH, and bile salts. In the same context, the high hydrophobicity and autoaggregation of fresh bee pollen lactobacillus strains are necessary characteristics for the bacterial adhesion to the host system, and its protection through biofilm formation over the host intestinal tissue, making these bacteria promising candidates for use as novel probiotics in the food and pharmaceutical industries [122]. In a recent Turkish study, another bacterial strain known as “fructophilic lactic acid bacteria (FLAB)” was isolated from fresh bee pollen and bee bread samples. FLAB are lactic acid bacteria (LAB) that prefer fructose over glucose as a carbon source and have been isolated from ecological fructose-rich niches including flowers and fruits, as well as the gastrointestinal tracts (GIT) of several fructose-based diet insects such as honeybees [123]. Several studies have shown the beneficial uses of probiotics in humans against type 2 diabetes, obesity, inflammation, tumors, allergy, metabolic disorders, and infectious diseases [124,125,126,127]. Due to this large bacterial population, bee pollen may have a promising role in the food and pharmaceutical industries.

### 4.4. Phenolic Profile and In Vitro Antioxidant Potential of Bee Pollen

#### 4.4.1. Volatile Compounds of Bee Pollen

The volatile content of bee pollen is rarely studied; indeed, two recent studies on samples from Lithuania, China, and Spain showed the presence of 42 different volatile compounds, mainly nonanal, dodecane, tridecane, hexane, 6-methyl-5-hepten2-one, methyl butanoic acid, limonene, and styrene [113,128]. These compounds are mainly found in flowers and participate in the attracting behavior of pollinators [129]. In other studies (from Greece and Poland), the results showed the presence of aldehydes, ketones, terpenoids, and minor amounts of furfural [115,130]. Bee pollen’s aromatic profile is related to the botanical and geographical origins as well as climatic conditions and bee species [28,113].

#### 4.4.2. Phenolic Profile of Bee Pollen

Bioactive compounds of bee pollen constitute an important quality criterion. Bee pollen as a natural product has gained great scientific interest due to its beneficial properties [131]. Interestingly, bioactive compounds may counteract the installation and/or development of different pathologies [132]. Therefore, the determination of the phenolic profile of bee pollen is considered the first step toward the standardization and prediction of the usefulness of this beehive product. The analysis of their composition revealed that the polyphenolic content presented an average of 3% to 5% of their composition, depending on the botanical origin of the bee pollen [133].

Phenolic acids represent an average of 0.19% of bee pollen, and their properties are mainly related to their structure [132]. Phenolic acids could be divided into benzoic acids, phenylacetic acid, and cinnamic acids, which are of the greatest interest and exert a good antioxidant activity as compared to the other phenolic acids groups. The main molecules of phenolic acids are chlorogenic acid, gallic acid, cinnamic acid, ferulic acid [103], hydroxycinnamic acid, and coumaric acid [134]. Flavonoids, on the other hand, are the most important group of polyphenols found in bee pollen with an average of 0.25% and 1.4% of its total composition; they are an excellent indicator of bee pollen quality [134]. Flavonoids are mostly found in bee pollen as glycosides (flavonoids associated with sugar units), with flavonol glycosides being the most abundant. However, the presence of glycoside bonds decreases the antioxidant activity of flavonols because of the steric effects, the reason why the content of bee pollen in free flavonoids is a good quality criterion [135,136]. The main flavonols identified in bee pollen are quercetin, kaempferol, and rutin. The main flavones are represented by apigenin, chrysin, and luteolin, and flavanones are represented by naringenin and pinocembrin, while genistein is the major isoflavone identified in bee pollen [132]. Additionally, resveratrol, the most important stilbene has been isolated by Ares et al. [137]. Table 2 summarizes the different isolated phenolic compounds with the different extraction techniques.

**Table 2 antioxidants-12-00557-t002:** Summary of the different methods used to determine the different active molecules of pollen.

Country	Floral Origin	Techniques	Phenolic Compounds Name	References
Turkey	ND	HPLC-PDA detector	gallic acid, 3,4-hydroxybenzoic acid, (+)-catechin1,2-dihydroxy-benzene, syringic acid, caffeic acid, rutin trihydrate, p-coumaric acid, trans-ferulic acid, apigenin 7 glucoside, resveratrol, quercetin, trans-cinnamic acid, naringenin, kaempferol, isorhamnetin	[131]
Turkey	*Castanea* spp.	HPLC-DAD	rosmarinic acid, vitexin, hyperoside, pinocembrin, trans-chalcone, apigenin, protocatechuic acid, galangin	[133]
Portugal	Monofloral: *Rubus* spp., *Cystisus* spp., *Quercus* spp., *Prunus* spp., *Leontondon* spp., *Cistus* spp., and *Trifolium* spp. Heterofloral: Castanea sativa and Echium spp. and ii) *Erica* spp., and *Eucalyptus* spp.	UHPLC-DAD-ESI-MS	coumaroyl quinic acid, myricetin-O-rutinoside, luteolin-O-dihexoside, quercetin-O-dihexoside, myricetin-O-hexoside, myricetin-O-(malonyl)rutinoside, isorhamnetin-O-dihexoside, quercetin-O-hexosyl-pentoside, quercetin-O-rutinoside isomer 1, quercetin-O-rutinoside isomer 2, luteolin-di-O-hexosyl-rhamosíde, quercetin-O-(malonyl)rutinoside, isorhamnetin-O-rutinoside, hydroxybenzoyl myricetin, quercetin-O-(malonyl)hexoside, quercetin derivative, quercetin-O-rhamnoside, isorhamnetin-O-(malonyl)hexoside isomer 1, luteolin-O-(malonyl)hexoside, myricetin, isorhamnetin-O-(malonyl)hexoside isomer 2, myricetin-O-dihydroferuloyl protocatechuic acid, myricetin-O-acetyl hydroxybenzoyl protocatechuic acid-isomer 1, myricetin-O-acetyl hydroxybenzoyl protocatechuic acid isomer 2, quercetin-O-acetyl hydroxybenzoyl protocatechuic acid isomer 1, myricetin-O-acetyl hydroxybenzoyl hydroxybenzoic acid isomer 2, quercetin-O-acetyl hydroxybenzoyl hydroxybenzoic acid isomer 1, quercetin-O-acetyl hydroxybenzoyl hydroxybenzoic acid isomer 2, O-dihydroxy benzoyl acetyl malonyl coumaric acid flavonoid derivative	[103]
China	*Rosa rugosa*	UPLC-ESI-QTOF-MS/MS	isorhamnetin 3-O-diglucoside, sorhamnetin-3-O-coumaroyl glucoside, isorhamnetin-3-O-6-O-acetyl-β-D-glucopyranosy, kaempferol-3-O-neohesperidoside, N′,N″,N‴-Tricaffeoyl spermidine, N′,N″,N‴-Dicaffeoyl p-coumaroyl spermidine, N′,N″,N‴-Di-p-coumaroyl caffeoyl spermidine, N′,N″,N‴-Tri-p-coumaroyl spermidine	[134]
Chile	*Brassica rapa* and *Eschscholzia californica*	HAPLC-DAD	syringic acid, coumaric acid, sinapic acid, ferulic acid, cinnamic acid, abscisic acid, catechin, myricetin, quercetin, apigenin, kaempferol, naringenin, rhamnetin	[135]
Brazil	*Eucalyptus marginata;* *Corymbia calophylla*	HPLC	gallic acid, 4-hydroxyphenylacetic acid, rutin, resveratrol, myricetin, quercetin-3-O-glucopyranoside, kaempferol-3-O-glucoside, kaempferol-3-O-rutinoside, naringenin, quercetin, phloretine, kaempferol	[136]
Morocco	*Coriandrum sativum*	HPLC/DAD/ESI-MSn	myricetin-3-O-rutinoside, quercetin-diglucoside, quercetin-3-O-rutinoside, kaempferol-3-O-rutinoside, isorhamnetin-3-O-rutinoside, isorhamnetin-O-pentosylhexoside, kaempferol-diglucuronide, isorhamnetin-3-O-glucoside, quercetin-3-O-rhamnoside, ellagic acid, N1-p-coumaroyl-N5, N10-dicaffeoylspermidinea, N1, N10-di-p-coumaroyl-N5-caffeoylspermidine, luteolin, quercetin-3-methyl-ether, N1, N5-di-p-coumaroyl-N10-caffeoylspermidine, N1, N5, N10-tri-pcoumaroylspermidine, N1, N5, N10-tri-pcoumaroylspermidine, N1, N5, N10-tri-pcoumaroylspermidine, N1, N5, N10-tri-pcoumaroylspermidine	[138]
Romania	*Hedera*, *Helianthus*, *Cistus*, *Cornus*, *Brassica*, *Gledistia*, *Hedysarum*, *Trifolium*, *Castanea*, *lamium*, *Magnolia*, *Fraxinus*, *Papaver*, *Crataegus*, *Prunus*, *Rubus*, *and Cordiandrum*	HPLC-DAD	gallic acid, protocatechuic acid, p-hydroxybenzoic acid, vanillic acid, caffeic acid, chlorogenic acid, p-coumaric acid, rosmarinic acid, myricetin, luteolin, quercetin, kaempferol	[139]
Italy	*Cistus ladanifer*, *Echium*, *Achillea, Quercus ilex*, *Rubus*, Pinaceae, *Filipendula*, *Trifolium incarnatum*, *Trifolium pratense*, *Trifolium repens*, *Prunus*, *Pyrus*, *Malus*, and *Oxalis*	UHPLC-ESI-QTOF-MS	cyanidin 3-O-xyloside/arabinoside, delphinidin 3-O-(60 ’-p-coumaroyl-glucoside), petunidin 3-O-arabinoside, pelargonidin 3-O-glucoside, delphinidin 3-O-glucoside, delphinidin 3-O-glucosyl-glucoside, delphinidin 3-O-rutinoside, cyanidin 3-O-sophoroside, naringin 60-malonate, naringin 40-O-glucoside, naringenin 7-O-glucoside, apigenin 7-O-(60′-malonyl-apiosyl-glucoside), tetramethylscutellarein, luteolin 7-O-glucuronide, apigenin 6-C-glucoside, kaempferol 3-O-glucuronide, quercetin 3-O-rutinoside, kamepferol 3,7-O-diglucoside, quercetin 3-O-galactoside 7-O-rhamnoside, quercetin 3-O-rhamnosyl-galactoside, kaempferol 3-O-sophoroside, 3,7-Dimethylquercetin, dihydroquercetin, formononetin, genistin, gallic acid ethyl ester, syringic acid, caffeic acid 4-O-glucoside, caffeoyl glucose, feruloyl glucose, caffeic acid, sesamol, hydroxytyrosol 4-O-glucoside, curcumin, and carnosic acid	[140]
Colombia	*Cistus ladanifer*; *Echium Achillea*; *Taraxacum*; *Carduus*; *Cirsium*; *Vicia*; *Quercus ilex*; *Rubus*; *Pinaceae*; *Filipendula*; *Trifolium incarnatum*, *Trifolium pratense*; *Trifolium repens*; *Prunus*, *Pyrus*; *Malus* and *Oxalis*	UHPLC-DAD	Caffeic acid, ferulic acid, S-N1,5,10-tri-ferulic acid isorhamnetin, kaempferol, luteolin, myricetin, p-coumaric acid, SP-N1,5,10,14-tetra-p-coumaric acid, pinobanskin, quercetin, spermidine, spermine, 4-methyl gallic acid, apigenin, amentoflavone, N1-caffeoyl-N5,10-di-p-coumaroyl-spermidine, and N1,10-di-pcoumaroyl-N5-caffeoyl-spermidine.	[44]

ND: Not determined.

#### 4.4.3. In Vitro Antioxidant Activity of Bee Pollen

Oxidative stress is involved in the development of several pathologies such as diabetes, Alzheimer’s, cancer, atherosclerotic, and other disorders [141]. The use of natural products such as beehive products including honey, pollen, royal jelly, and propolis as a source of antioxidant molecules has been supported and suggested to protect human cells from the effects of oxidative stress by numerous scientific studies [142]. Moreover, the antioxidant activities of bee pollen have been evaluated in several works using well-known techniques such as DPPH, ABTS, β- carotene, FRAP, CUPRAC, NO, and TAC assays [132,138,140,143,144,145,146,147,148,149,150,151,152,153,154,155,156,157,158,159,160,161,162,163,164,165,166,167,168,169,170]. Table 3 summarizes all studies that have evaluated the antioxidant activity of bee pollen. It presents the pollen origin, the type of extract, the methods used, and the main results. Indeed, El Ghouizi et al. [138] evaluated in vitro the antioxidant activity of the aqueous extract of Moroccan fresh bee pollen and revealed an important scavenging capacity against DPPH and FRAP with IC_50_ values of 0.39 ± 0.13 mg/mL and 0.54 ± 0.53 mg/mL, respectively.

In Brazil, several authors have investigated bee pollen samples from several botanical and geographical origins for their antioxidant proprieties using four antioxidant assays (DPPH, ORAC, β-carotene, and FRAP) and the results revealed that the bee pollen extracts exhibited important antioxidant activity in all tests with a significant difference among them, and a potential correlation between this activity and polyphenolic composition, which in turn varied depending on the geographical and botanical origins of the plant visited by the bees [145,150,151,168,171].

In Egypt, authors have investigated the antioxidant activity of ethanol, methanol, petroleum ether, dichloromethane, and ethyl acetate extracts of bee pollen samples from monofloral sources (*Zea mays* and *Trifolium alexandrinum* L) by two in vitro methods (ABTS and DPPH). The results showed an interesting anti-DPPH activity of ethanol extract with a percentage activity of 90%, while the methanol extract revealed a strong activity using the ABTS test with a percentage activity of 76.51%. Based on the findings, the high antioxidant activity of ethanol extract could be related to its major compounds including catechin, quercetin, and caffeic and gallic acid [144,162]. These results are confirmed by studies on Chinese bee pollen. In those works, authors investigated the antioxidant capacity of aqueous, ethanol, and methanol extracts using DPPH, ABTS, superoxide-scavenging activity, and reducing power and showed that ethanol extract of bee pollen has a good anti-DPPH effect with IC_50_ = 1.28 ± 0.03 mg/mL, and an important anti-ABTS (1.06 ± 0.02 mmol TE g^−1^) and reduction power activity (70.55%). The methanol extract showed also an antioxidant effect with an anti-DPPH IC_50_ value of 1.72 mg/mL, and IC_50_ = 3.48 mg/mL for superoxide-scavenging activity [157,161,169]. Methanol extracts of twenty-two bee pollen samples from different floral origins in Portugal also showed an interesting antioxidant effect, which manifests in scavenger activity of the free DPPH• and β-carotene in bleaching assays with mean values of 3.0 ± 0.7 mg/mL and 4.6 mg/mL, respectively [155].

On the other hand, several authors have reported the antioxidant activity of Turkish bee pollen extracts based on different geographical origins [160,167], bee races (*Apis mellifera caucasica*, *Apis mellifera anatoliaca*, *Apis mellifera syriaca*, and *Apis mellifera carnica*) [166], and extraction methods as well as the storage conditions [152]. The authors showed important antiradical activity of bee pollen which is highly affected by chemical composition, plant origins, geographical origin, and storage conditions.

Other research works reported a variable antioxidant effect of different bee pollen extracts including those from Algeria, Greece, Italy, Korea, Spain, Slovakia, Malaysia, Serbia, Thailand, and Poland, and this variability can be significantly related to a variety of botanical and geographical origins [132,140,143,147,148,153,154,156,158,159,163,164,165,170].

## 5. Therapeutic Properties of Bee Pollen against Oxidative Stress-Related Diseases

The use of bee pollen in traditional medicine dates to ancient times and is attested by books of Arab and Jewish doctors such as Ibn al-Beithar and Maimonides in the early 1100s, where they described bee pollen as an aphrodisiac, sedative, and effective for the stomach, the heart, and intestines [172,173].

### 5.1. Antioxidative Properties

Oxidative stress is caused by an imbalance between free radical production and antioxidant defense systems, resulting in an accumulation of reactive oxygen species (ROS). In turn, ROS interact with various cytoplasmic components such as proteins, membrane lipids, and DNA [174]. As a result, ROS induce serious cell damage and participate in the development of many chronic illnesses such as diabetes and associated complications, arthritis, Parkinson, and Alzheimer’s [175]. Bee pollen is one of the natural antioxidant-rich products mainly used against oxidative stress and related pathologies. In this context, Kawther and coworkers have proved that the administration of bee pollen extract (250 mg/kg b.w) attenuated oxidative stress induced by protein [176]. It has been demonstrated that the anti-oxidative activity of bee pollen is attributed to its content of secondary metabolites including, vitamin E, vitamin C, carotenoids, and phenolic compounds [149]. Thanks to its phenolic hydroxyl group, the flavonoids present in bee pollen can scavenge ROS and free radicals, and inactivate electrophiles [164]. As reported in Table 4, several studies have reported the antioxidative properties of various phenolic compounds and their involved mechanisms. Pari et al. reported that the sub-chronic administration of Caffeic acid (6 mg/kg b.w) improved the oxidative stress caused by alcohol-induced toxicity in rats by increasing non-enzymic antioxidant defense systems, and by preventing lipid peroxidation [177]. Likewise, cinnamic acid occurs in the antioxidative process by modulating lipid metabolism and boosting GSH, SOD, and CAT enzyme activities as well as scavenging and decreasing ROS production [178]. In addition, the oral administration of rutin at a dose of 50 and 100 mg/kg/b.w for 20 days enhances the production of antioxidant enzymes, decreases serum toxicity markers, and downregulates COX, 2p38-, MAPK, i-NOS, and the NF-κB signaling pathway [179]. Furthermore, a recent study showed that quercetin weakened oxidative stress and decreased the expression of TNFα, IL-1β, and IL-6 [180]. Similarly, treatment with luteolin minimized the oxidative stress through multiple mechanisms: (a) up-regulation of the Nrf-2 pathway, (b) enhancement of HO-1 expression, (c) increase in GSH, SOD, and GPX activities, and (d) decrease in MDA levels [181]. Pinocembrin decreased oxidative stress, apoptotic, and inflammatory markers [182] Figure 1.

### 5.2. Antidiabetic and Anti-Hyperglycemic Properties

Diabetes mellitus is an endocrine disorder characterized either by insufficient insulin secretion and/or its defective utilization [183]. The antidiabetic/anti-hyperglycemic activity of bee pollen has been previously studied. According to Nema et al., bee pollen administration at 100 mg/kg body weight/day for 4 weeks lowered blood glucose and prevented pituitary–testicular axis dysfunction [6]. Furthermore, bee pollen exhibited a potent anti-hyperglycemic activity in patients with insulin-independent diabetes mellitus (T2DM) [184]. Moreover, thanks to its bioactive constituents, bee pollen exerts its anti-hyperglycemic effect by modulating glucose uptake and inhibiting α-amylase and β-glucosidase activities, leading to the management of diabetes and its serious complications [185]. The anti-diabetic properties of many individual phenolic compounds present in bee pollen have already been investigated. Adisakwattana and coworkers enunciated that cinnamic acid administered orally at a dose of 50 mg/kg/day for 5 weeks stimulates insulin and adiponectin secretions, increases hepatic glycolysis, improves glucose uptake, potentiates pancreatic β-cell functionality, and decreases protein glycation [186]. Rutin regulates glycemia and ensures its anti-diabetic effect through the inhibition of the polyol signaling pathway as well as via the modulation of lipid metabolism and the prevention of lipid peroxidation [187]. Apigenin facilitates and enhances GLUT4 translocation in skeletal muscles either by up-regulating the AMP-activated protein kinase pathway or by activating the insulin signaling pathway, which leads to glucose uptake and thus hypoglycemia. A paper published by Alkhalidy et al. explored the anti-diabetic effect of kaempferol against streptozotocin-induced diabetes in rats and found that chronic administration of kaempferol (50 mg/kg-b.w) reduced hepatic glucose production, increased hexokinase activity, decreased hepatic pyruvate carboxylase activity, and inhibited the gluconeogenesis pathway [188], as shown in Figure 1.

### 5.3. Hepatoprotective Properties

Bee pollen extract has been found to possess a potent hepato-protective effect. Cheng and coworkers reported that bee pollen extract administration increased tissue catalase SOD and GSH-Px activity, and prevents liver histological changes induced by carbon tetrachloride treatment in mice. This suggests the potential role of bee pollen in preventing hepatocellular changes associated with exposure to xenobiotics. The hepatoprotective capacity of bee pollen is largely attributed to its rich content of natural antioxidants such as phenols and flavonoids [189]. The hepatoprotective ability of phenolic compounds has been explored in numerous studies (Table 3). Malayeri et al. showed that the co-administration of a single dose of naringenin (50 mg/kg/b.w) boosted enzymatic and non-enzymatic antioxidant activities, and reduced NO, TNF-α, and IL-6 levels [190]. Yang and coworkers indicated that ferulic acid exhibits its protective role against CCL4-caused acute oxidative liver damage in rats via the up-regulation of p-JNK, p-p38 MAPK, and Bcl-2 signaling pathways and thus, decreased the expression of pro-inflammatory mediators of hepatic-toxicity TNF-α and IL-1β [191]. Owumi et al. recently emphasized that protocatechuic acid protects against methotrexate-induced liver dysfunction via enhancing enzyme antioxidant defense mechanisms and decreasing oxidative stress and free radical production, which was confirmed by biochemical analysis and histopathological investigations [192]. Vanderson et al. demonstrated that caffeic acid treatment improves oxidative stress and kidney dysfunction mediated by ethanol in a rat model. This was attributed to the down-regulation of CYP2E1 and the protection of DNA against oxidative damage [193]. Ebrahimi and coworkers proved that ellagic acid treatment reduced oxidative damage and liver ultrastructure changes in methotrexate-induced mitochondrial dysfunction and liver toxicity in rats [194] (see Figure 1).

### 5.4. Nephroprotective Properties

Owing to its rich content of various bioactive molecules, bee pollen has been found to have a potent nephroprotective activity. In a rat model, it was reported that bee pollen extract improved biochemical parameters (creatinine and bilirubin), increased the antioxidant defense system (SOD, CAT, and GSH), lowered oxidative stress biomarkers (MDA and iNOS), and prevented kidney histological effects induced by cisplatin. Intraperitoneal administration of apigenin reduced COXI, COXII, and MDA levels, and increased kidney GSH levels [195]. Recently, Owumi et al. reported that protocatechuic acid exhibits its reno-protective activity by increasing the activity of antioxidant enzymes (SOD, CAT, GSH, and GPX) and decreasing the renal (RNOS and LPO) levels, as well as reducing the inflammation biomarkers NO, TNF-α, and IL-1β levels in renal tissue [192]. Another study found that naringenin (100 mg/kg/b.w) reduced oxidative stress and prevented lipid peroxidation in rats after cyclosporine treatment [196]. According to Chowdhury and colleagues, ferulic acid prevented hyperglycemia-induced kidney damage and oxidative stress in rats; this was related to the modification of AGEs, MAPKs (p38 and JNK), and the NF-B signaling pathways by this acid [197]. Mohammed and coworkers evidenced that ellagic acid stimulated the expression of SIRT1, decreased P53 protein levels, reduced ROS production, and enhanced enzymatic and non-enzymatic antioxidant systems [198]. This was demonstrated through improved biochemical values (creatinine, urea, and uric acid), kidney histopathological tissue, and renal biomarker stress (MDA, GSH, CAT). Another study showed that pinocembrin treatment mitigates gentamicin-induced inflammation and renal toxicity via the modulation of Nrf2/HO-1 and NQO1 pathways [199]. This could suggest its potent ability to reduce oxidative stress and inflammation-related nephrocellular dysfunctions. Shanmugam et al. reported that oral administration of Kaempferol (100 mg/kg/day/b.w) exerts its nephroprotective action by inhibiting RhoA/Rho kinase-mediated inflammatory pathway [200] Figure 1.

### 5.5. Anti-Inflammatory Properties

Numerous scientific studies have indicated that bee pollen has a potent anti-inflammatory impact. Indeed, it has been shown that flavonoids and phenolic acids play a major role in the anti-inflammatory activity of bee pollen extracts. As indicated in Table 3, several individual phenolic components have shown anti-inflammatory effects through different signaling pathways. Indeed, phenolic acids including caffeic acid, ferulic acid, and cinnamic acid are documented as potent inhibitors of tumor necrosis factor (TNF) and thus down-regulation of the nuclear factor NF-κB pathway (proinflammatory signaling pathway) [201,202,203]. In addition, ellagic acid has been found to inhibit nitric oxide (NO), TNF-α, and IL-6, and induce the down-regulation of cyclooxygenase II (COX-2) and prostaglandin E2 (PGE2) [204]. The anti-inflammatory activity of bee pollen is also linked to Galangin, chrysin, quercetin, resveratrol, kaempferol, and other flavonoid molecules. Choi and coworkers have shown that galangin inhibits the expression of iNOS, COX-2, and the release of pro-inflammatory cytokines such as IL-1β and TNF-α [205]. In the same context, the findings of Li, Zhipeng, et al. showed that chrysin improves inflammatory reaction through the inhibition of NO, prostaglandin E2, and the NF-κB signaling pathway [206]. Quercetin and resveratrol exhibit their anti-inflammatory actvity via the down-regulation of the NF-κB pathway and the inhibition of COX-1 and COX-2 activities [207,208]. Park and coworkers reported that Kaempferol ensures its protective effect in aged kidney tissues via the suppression of pro-inflammatory cytokines (IL-1β, TNF-α, IL-18, and IL-6) [209]. These data revealed that kaempferol could have the ability to attenuate age-related chronic inflammatory reactions Figure 1.

**Table 4 antioxidants-12-00557-t004:** Summary of the different pharmacological properties of different phenolic compounds found in bee pollen.

Molecules	Dosage, Route, and Exposure Duration	Pharmacological Properties	Involved Mechanisms	References
Caffeic acid	6 mg/kg/day, orally for 45 days.	Anti-oxidative properties	↑Non-enzymic antioxidants, ↓lipid peroxidation, and ↓TBARS level.	[177]
Cinnamic acid	20 mg/kg/day, i.p for 40 days.		↓lipid peroxidation, ↓ROS production ↑GSH, ↑SOD, and ↑CAT levels.	[178]
Ferulic acid	25 mg/kg/day, orally for 10 days.		↓lipid peroxidation, ↓ROS levels, and ↓N-acetyl-β-glucosminidase activity.	[210]
Ellagic acid	10 and 30 mg/kg/day for 30 days.		Enhances the concentration of enzymatic antioxidant levels (SOD, CAT, and GPx), and ↓ MDA, ↓TNF-α, and ↓IL-1β.	[211]
Quercetin	50 mg/kg/day, i.p for 21 days.		Increases GSH level, SOD, GR, G, P, and CAT activity, and decreases the expression of TNFα, IL-1β, and IL-6.	[180]
Kaempferol	100 mg/ kg/day, i.p for 6 weeks.		Inhibits the activity of ASK1/MAPK signaling pathways (JNK1/2 and p38).	[212]
Galangin	8 mg/kg/day, i.p for 45 days.		↓lipid peroxidation, ↑enzymatic and non-enzymatic antioxidants.	[213]
Chrysin	30 mg/kg/day, orally, for 14 days.		↑GSH, ↓TBARS, ↓XO, and ↓NADPH levels	[214]
Protocatechuic acid	100 mg/kg/day, i.p for 7 days.		Prevents lipid peroxidation and the formation of NO, and enhances antioxidant enzymes.	[215]
Apigenin	0.625, 1.25, and 2.5 mg/mL.		↓oxidative stress, GSH level ↑SOD activity, ↓IL-6, and ↓NF-κB levels.	[216]
Luteolin	100 and 200 mg/kg/ day, orally for 28 days.		↓MDA, ↑GSH, ↑SOD ↑GPX ↑Nrf2, and ↑HO-1 Expressions.	[181]
Rutin	50 and 100 mg/kg/day, orally for 20 days.		↑Production of antioxidant enzymes, ↓serum toxicity markers, and downregulation of (COX, 2p38-, MAPK, i-NOS, and NF-κB).	[179]
Naringenin	50 mg/kg/day, orally for 8 weeks.		Minimizes oxidative stress and enhances CAT, SOD GSH, and GPx levels.	[217]
Pinocembrin	10 mg/kg/day, orally for 7 days.		Decreases oxidative stress, and apoptotic and inflammatory markers.	[182]
Caffeic acid	100 mg/kg/day, orally for 4 weeks.	Antidiabetic and anti-hyperglycemic properties	IL-6, ↓ IL-1β, ↓ TNF-α, ↓ MCP-1, ↓HbA1c, ↓ UGA, ↓ sorbitol, ↓ fructose, and ↑AMPKα2.	[218]
Ferulic acid	10 mg/kg/day, orally for 15 days.		Down-regulation of NF- κB pathway.	[219]
Cinnamic acid	50 mg/kg/day, orally for 5 weeks.		↑insulin secretion, ↑hepatic glycolysis, ↑adiponectin secretion ↑glucose uptake, ↑pancreatic β-cell functionality, and ↓protein glycation.	[186]
Ellagic acid	250 mg/kg/day, orally for 28 days.		↑ insulin secretion, ↑β-cell number, ↑plasma total antioxidants, and ↑glucose intolerance.	[220]
Quercetin	10 and 30 mg/kg/day, i.p for 14 days.		↑GLUTs, ↑IR-P, ↑GLUT4, ↑Glucose uptake, ↑pancreatic cell-β generation, ↑glucokinase activity, ↓α-glucosidase activity.	[221]
Kaempferol	50 mg/kg/day, orally for 12 weeks. 200 mg/kg/day, orally for 14 days.		↓hepatic glucose production, ↑hexokinase activity, ↓hepatic pyruvate carboxylase activity, and gluconeogenesis. ↑GLP-1 and insulin release, ↑ cAMP, and Ca2+ intracellular levels.	[188,222]
Galangin	4, 8, and 16 mg/kg/day, orally for 45 days.		Inhibition of DPP-4, ↓oxidative stress, and ↑antioxidant status.	[213]
Chrysin	40 mg/kg/day, orally for 16 weeks.		Inhibition of the TNF-α pathway, ↓secretion of pro-inflammatory cytokines, and ↓glucose and lipid peroxidation levels.	[223]
Protocatechuic acid	50 and 100 mg/kg/day, orally for 7 days.		↑insulin sensitivity, ↓insulin resistance, ↓gluconeogenesis, and ↑glucose uptake.	[224]
Apigenin	1.5 mg/kg/day, i.p for 28 days.		Enhances GLUT4 translocation.	[225]
Luteolin	10 mg/kg/day, orally for 24 weeks.		Reduces oxidative stress and inhibits the STAT3 pathway.	[226]
Rutin	90 mg/kg/day, orally for 10 weeks.		Inhibition of polyol pathway, oxidative stress, and lipid peroxidation.	[187]
Naringenin	50 and 100 mg/kg/day, orally for 6 weeks.		Improvement of glucose and lipid metabolism, and ↓insulin resistance.	[227]
Pinocembrin	50 mg/ kg/day, orally for 10 days.		↓ NF-κB and TNF-α levels.	[228]
Resveratrol	12 mg/kg/day, orally for 1 5 days.		Down-regulation of NF- κB pathway.	[219]
Caffeic acid	100 mg/kg/day, orally for 4 days.	Hepato-protective properties	Downregulation of CYP2E1 and the protection of DNA against oxidative damage.	[193]
Cinnamic acid	20 mg/kg/day, orally for 10 days.		↓NF-kB and ↓iNOS activities.	[229]
Ellagic acid	5 and 10 mg/kg/day, orally for 10 days		Up-Regulation of Nrf2 and HO-1 expression and inhibition of NF-κB signaling pathway.	[194]
Quercetin	20, 40, and 80 mg/kg/day, orally for 7 days.		Modulation of the expression of nuclear orphan receptors (CAR, PXR) and cytochrome P450 enzymes (CYP1A2, CYP2E1, CYP2D22, CYP3A11).	[230]
Kaempferol	20 mg/kg, twice a day, orally for 28 days.		↓CYP2E1 activity and ↓ROS production.	[231]
Galangin	15, 3,0, and 60 mg/kg/day, orally for 15 days.		Activation of Nrf2 and HO-1 signaling pathway.	[232]
Chrysin	25 or 50 mg/kg, orally for 6 days.		Decreases the expression of COX-2, iNOS.	[233]
Ferulic acid	25, 50, and 100 mg/kg/day, orally for 7 days.		↓ the expression TNF-α and IL-1β, upregulation of p-JNK, p-p38 MAPK, and Bcl-2.	[191]
Protocatechuic acid	25 and 50 mg/kg/day, i.p for 7 days.		↓ oxidant species ↑antioxidant enzymes	[192]
Apigenin	10 mg/kg/day, orally for 3 weeks.		Enhances antioxidant defense mechanisms and decreases lipid peroxidation.	[234]
Luteolin	100 mg/kg/day, i.p for 7 days.		Modulation of Nrf2/HO-1 pathway and ↓oxidative stress.	[235]
Rutin	20 mg/kg/day, orally for 15 days.		↑Antioxidant profile and regulation of Na+/K+ ATPase activity.	[236]
Naringenin	50 mg/kg/day, orally for 10 days.		↑the enzymatic and non-enzymatic antioxidant levels, ↓NO, TNF-α, and IL-6 levels.	[190]
Pinocembrin	50 and 75 mg/kg/day, i.p for 10 days.		Modulation of Nrf2/HO-1 and NQO1 pathways.	[199]
Resveratrol	50 and 100 mg/kg/day, orally for 28 days		Modulation of SIRT1 and p53 pathways.	[237]
Caffeic acid	100 mg/kg/day, orally for 14 days.	Nephroprotective properties	Enhances the antioxidant defense system and reduces lipid peroxidation.	[238]
Ferulic acid	50 mg/kg/day, orally for 8 weeks.		Modulation of AGEs, MAPKs (p38 and JNK), and NF-κB pathways, and ↓oxidative stress.	[197]
Cinnamic acid	50 mg/kg /day, orally for 7 days.		antioxidant expression GSH levels, SOD, CAT, and GPx activities.	[239]
Ellagic acid	10 mg/kg/day, orally for 30 days.		Stimulates the expression of SIRT1, ↓P53 protein level, ↓ROS production, and ↑enzymatic and non-enzymatic antioxidant system.	[198]
Quercetin	10 mg/kg/day, i.p for 10 weeks.		↑antioxidant expression and ↓lipid peroxidation.	[240]
Kaempferol	100 mg/kg/day, orally for 28 days.		Inhibits RhoA/Rho Kinase mediated inflammatory pathway.	[241]
Chrysin	30 and 100 mg/kg, ip for 26 days.		↑iNOS and PKC Levels, and ↓AGEs and RAGE.	[242]
Protocatechuic acid	25 and 50 mg/kg/day, i.p for 7 days.		↓ oxidant species ↑antioxidant enzymes.	[192]
Apigenin	3 mg/kg/day, i.p for 7 days.		Reduces COXI and COXII, MDA levels and increases GSH level.	[195]
Luteolin	10 and 20 mg/kg/day, orally for 4 weeks.		Inhibition of RIP140/NF-κB pathway.	[243]
Rutin	100 mg/kg/day, orally for 14 days.		Suppresses NF-κB activation and TGF-β1/Smad3 signaling.	[244]
Naringenin	100 mg/kg/day, orally for 45 days.		↓ oxidative stress and lipid peroxidation levels.	[196]
Pinocembrin	50 and 75 mg/kg/day, i.p for 10 days.		Modulation of Nrf2/HO-1 and NQO1 pathways.	[199]
Resveratrol	20 mg/kg/day, orally for 40 weeks.		Modulation of the NF-κB signaling pathway.	[245]
Caffeic acid	50 mg/kg/day, orally for 21 days.	Anti-inflammatory properties	inhibition of NO, prostaglandin E2, and NF-κB signaling pathways.	[201]
Ferulic acid	100 mg/kg/day, orally for 6 weeks.		Inhibition of NADPH oxidase and NF-κB pathway.	[202]
Cinnamic acid	60 mg/kg/day, orally for 21 days.		Down-regulation of the NLRP3, NF- κB, and ASK1/MAPK signaling pathways.	[203]
Ellagic acid	1, 3, 10, and 30 mg/kg, i.p for 5 h.		Suppression of NF-κB pathway and NO, TNF-α, IL-6, COX-2 activity, and PGE2.	[246]
Quercetin	1 mg/kg/day, orally for 15 days.		Down-regulation of the NF-κB pathway.	[208]
Kaempferol	2 and 4 mg/kg/day for 10 days.		Decreases the synthesis of IL-1β, TNF-α, IL-18, and IL-6.	[209]
Galangin	50 mg/kg per day, orally for 4 days.		Inhibits the expression of iNOS, COX-2, and pro-inflammatory cytokines.	[205]
Chrysin	40 mg/kg/day, orally for 16 weeks.		inhibition of NO, prostaglandin E2, and NF-κB signaling pathways.	[206]
Protocatechuic acid	20 mg/kg/day, orally for 8 weeks.		↓IL-1β, ↓IL-6, and ↓TNF-α synthesis pathways.	[247]
Apigenin	20 and 40 mg/kg/day, orally for 28 days.		↓TNF-α and IL-6 production.	[248]
Luteolin	100 mg/kg, i.p for 6 h.		↑HO-1 expression, ↑IL-10, ↓TNF-α, and ↓IL-6 levels.	[249]
Rutin	30 mg/kg/day, orally for 14 days.		Inhibition of p38-MAPK pathway.	[250]
Naringenin	5, 10, and 20 mg/kg/day, for 16 days.		Up-regulation of Nrf-2/HO-1pathway and ↓NF-kB mRNA expression.	[251]
Pinocembrin	50 mg/kg/day, i.p for 24 days.		Down-regulation of NF-kB pathway.	[252]
Resveratrol	10 or 50 mg/kg/day, orally for 28 days.		Inhibition of COX-1 and COX-2 activities	[207]

↑ represent increases; ↓ represent decreases.

### 5.6. Other Beneficial Effects of Bee Pollen

Bee pollen has a broad spectrum of pharmacological effects and provides a promising area for researchers interested in the therapeutic effects of natural products, particularly hive products. The rich composition of probiotics, proteins, macro-, and micronutrients in bee pollen has been related to its positive effect on morphological development (thickness of epithelium) and functioning (absorption) of the small intestine, leading to the proper functioning of the gastrointestinal tract [217,253].

Bee pollen is also known to modulate the secretory activity (release of IGF-I growth factor and progesterone and estradiol steroid hormones) and apoptotic activity of the ovary in rats. In postmenopausal women with breast cancer, bee pollen can improve menopausal-related symptoms when used in association with antihormonal treatment; it is also beneficial for women who suffer from post-menopausal disorders [254].

Studies have demonstrated the positive effects of the phenolic and probiotic content of bee pollen on preventing metabolic syndrome by reducing body and liver weight gain, decreasing fasting blood glucose, and lipid accumulation in serum and liver, which can be explained through the regulation of intestinal microbiota [255].

In many cultures, bee pollen has long been used by women to maintain their beauty and whiten their skin. Since more than 70% of bee pollen composition is active, (proteins, carbohydrates, lipids/fatty acids, phenolic compounds, and vitamins), the cosmeceutical properties of bee pollen in the laboratory have been studied and researchers have demonstrated that pollen can boost protective mechanisms against skin aging (polyphenols, vitamin E, C), skin dryness (sugars and fatty acids), ultraviolet radiation (carotenoids), oxidative damage (polyphenols), and inflammation and melanogenesis, which are involved in human skin damage [256,257]. These scientific pieces of evidence are turning cosmetologists’ attention toward introducing bee pollen into their beauty products and formulations, and guaranteeing better quality and functionality.

## 6. Conclusions

As has been shown so far, pollen grains are microscopic vegetal cells produced and dispersed during the process of plant reproduction. In the hive, pollen grains transform into bee bread through the fermentation process and become accessible for human consumption because of their complete composition of macronutrients such as proteins, carbohydrates, lipids, and micronutrients such as minerals, vitamins, and phenolic compounds. This diversified composition affords a wide range of pharmacological and biological properties to the bee pollen. Nonetheless, due to the significant variability of its composition, which is affected by several factors, bee pollen application in phytomedicine remains quite limited. On this basis, scientists and professionals should pay closer attention to a few key aspects: (i) standardization should be expanded to include the phenolic composition and nutritional value of different types of bee-collected pollen, especially monofloral pollen; (ii) more quality-control research is needed to encourage beekeepers to produce clean, safe, and economically valuable bee pollen; (iii) as bee pollen is partially digested by human digestive enzymes, more pharmacological and biochemical studies are necessary to enhance the bioavailability of bee pollen bioactive compounds and capitalize on bee pollen’s biological importance; (iv) considering the techno-functional value of bee pollen as a superfood, it can be potentially used as a good ingredient in the food and pharmaceutical industries for the production of novel bee pollen-enriched food products or dietary supplements; (v) eventually, considering bee pollen’s techno-functional value and biological properties, more clinical trials should be conducted to investigate the beneficial effect of this superfood on human health and to encourage the food and pharmaceutical industries to develop and manufacture novel bee pollen-enriched food products and dietary supplements.

## Figures and Tables

**Figure 1 antioxidants-12-00557-f001:**
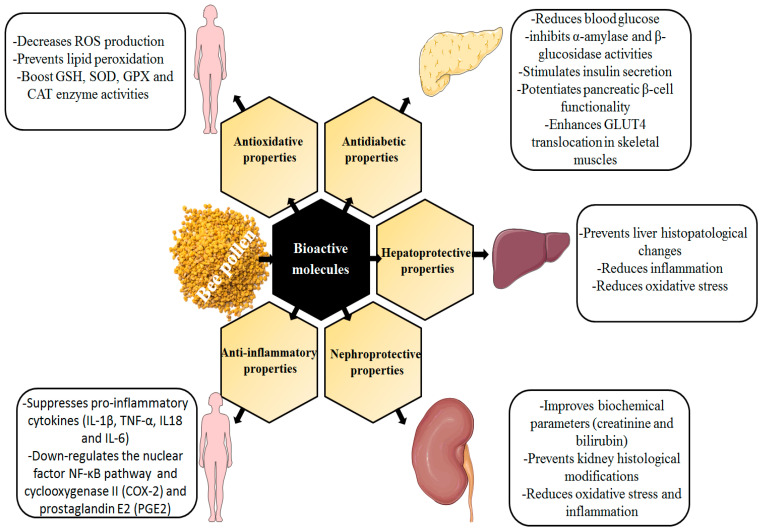
Schematic presentation of the therapeutic properties of bee pollen.

**Table 1 antioxidants-12-00557-t001:** Summary of the different vitamins isolated from bee pollen.

Identified Vitamins	Apiaries	Floral Origin	Isolation Methods	References
A, B1, B2, B5, B6, B7, B12, C, E, K2	Turkey	monofloral bee pollen of *Rhododendron ponticum*	HPLC-FLD HPLC-UV	[102]
β-Carotene Vit. C	Portugal	Polyfloral bee pollen of: *Rubus* spp.; *Castanea*; *sativa*; *Cytisus* spp.; *Quercus* spp.; *Echium* spp.; *Prunus* spp.; *Leontondon* spp.; *Eucalyptus* spp.; *Erica* spp.; *Cistus* spp.; *Trifolium* spp.	(NH4)2SO4 for β-Carotene AOAC for vitamin-C	[103]
B2, B3, B6, B9	Italy	Polyfloral bee pollen of: *Prunu*; *Erica*; *Brassicaceae*; *Rubus*; *Viburnum* *Viburnum*; *Trifolium pratense*; *Asteraceae T.*; *Eucalyptus*; *Rosa* spp.	Fluorescence spectroscopy (Bulk analysis)	[104]
B1, B2, B6	Brazil	Polyfloral bee pollen of: *Arecaceae*; *Cecropia*; *Cestrum*; *Cyperaceae*; *Eucalyptus*; *Ilex*; *Myrcia*; *Piper*; *Vernonia*; *Trema*	HPLC	[105]
B3 (Niacin); B6 (Pyridoxine) B9 (Folic acid) B12 (Cobalamin)	Saudi Arabia	Monofloral bee pollen of: *alfalfa*; date palm; *rape*; summer squash; sunflower	HPLC	[75]
C; E; Provit. A (β-carotene)	Brazil	ND	Vit C: AOAC Vit E: HPLC β-carotene: OCC	[106]

ND: Not determined.

**Table 3 antioxidants-12-00557-t003:** Summary of the different studies of the antioxidant activity of bee pollen.

Country	Botanical Origin	Extracts	Used Methods	Key Results	References
Morocco	*Coriandrum sativum*	Aqueous extract	DPPH	IC_50_ = 0.39 ± 0.13 mg/mL	[138]
			Ferric reducing power	IC_50_ = 0.54 ± 0.53 mg/mL	
			Total antioxidant capacity	56.92 ± 0.21 mg AAE/g	
Algeria	Monofloral samples: *wild carrot*, *rosemary*, and *eucalyptus*	Methanolic extract	Molybdate ion reduction Assay	101.58 ugGAE/g	[165]
Brazil	Monofloral: *Brassica genus*; *Brassica rapa*; *Astrocaryum*; *Aculeatissimum*; *Cocos nucifera*; *Myrcia*; *Alternanthera*; *M. scabrella*; *Eucalyptus*; *Coffea*; *M. scabrella*; *M. verrucosa*; *Eupatorium*; *Syagrus*; *A. aculeatissimum*; *Eupatorium*; *Myrcia*; *Cecropia*; *Myrcia*; *Alternanthera*; *M. caesalpiniifolia*; *Montanoa*; *Asteraceae*; *C. nucifera*; *Machaerium*; *M. caesalpiniifolia*; *Myrcia*; *Anadenanthera*; *Cecropia*; *Schinus*; *Ilex*; *Ricinus*	Ethanolic extract	DPPH	140 ± 5 mmol TE/g	[150]
			ORAC	563 ± 15 mmol TE/g	
Brazil	*Mimosa misera*, *Mimosa caesalpinifolia*, *Eythrina velutina*, *Ziziphus lotus*, *Prosopis juliflora*, *Mimosa tenuiflora*, *Piptadenia macrocarpa*, *Cautarea hexandra*, *Hyptis suavelens*, *Cautarea hexandra*, and *Maytenus rigida*	Ethanolic extract	β-carotene bleaching method	Antioxidant activity = 83.3%	[168]
Brazil	Heterofloral: *Arecaceae*, *Asteraceae baccharis*, and *Asteraceae eupatorium*	Hydroethanolic extract	FRAP	131.47 ± 75.08 mg GA eq/g	[145]
			DPPH	% inhibition = 72.46 ± 5.25%	
Brazil	*Arecaceae*; *Asteraceae baccharis*; *Asteraceae eupatorium*; *Brassicaceae*	Lyophilized extract	ABTS	120.10 ± 0.21 mmol TEAC/g	[145]
			DPPH	Antioxidant activity = 54.42 ± 0.23%	
			FRAP	60.64 ± 0.63 mmol of Fe2þ/g	
			β-carotene/linoleic acid Assay	Antioxidant activity = 91.93 ± 0.22%	
Brazil	ND	Hydroethanolic extract	DPPH	EC_50_ = 0.86 mg/mL	[151]
			FRAP	123.4 mgGAEq.100 g^−1^	
			β-carotene/linoleic acid Assay	Antioxidant activity = 83.3%	
Chile	*Tilia Tuan* Szyszyl	Aqueous extract	DPPH	IC_50_ = 2.36 mg/mL	[161]
			Superoxide-scavenging activity	IC_50_ = 2.29 mg/mL	
		Methanolic extract	DPPH	IC_50_ = 1.72 mg/mL	
			Superoxide-scavenging activity	IC_50_ = 3.48 mg/mL	
China	*Agastache rugosatache* rugosa *Brassica napus* L. *Camellia japonica* L. *Crataegus pinnatifi* *Dendranthema indicum* L. *Fagopyrum esculentum moench* *Helianthus annuus* L. *Nelumbo nucifera Gaertn.* *Phellodendron amurensis* *Prunus armeniaca* *Prunus persica* L. *Rosa rugosa Thunb.* *Schisandra chinensis* *Taraxacum mongolicum*	Hydroethanolic extract	ABTS	1.06 ± 0.02 mmol TE g^−1^	[169]
			DPPH	IC_50_ = 1.28 ± 0.03 mg/mL	
			Reducing power	Antioxidant activity = 70.55 ± 0.00%	
China	*Lotus uligionosus*, *Escallonia rubra*	Aqueous extract	DPPH	119.9 eq/g	[157]
			Reducing power	69.5 eq/g	
Egypt	*Trifolium alexandrinum* L.	Ethanolic extract	DPPH	Antioxidant activity = 90%	[144]
		Petroleum ether	DPPH	Antioxidant activity = 75%	
		Dichloromethane	DPPH	Antioxidant activity = 63%	
		Ethyl acetate	DPPH	Antioxidant activity = 79%	
Egypt	*Zea mays*	Methanolic extract	DPPH	Antioxidant activity = 59%	[162]
			ABTS	Antioxidant activity = 76.51%	
Greece	Monofloral sample: *Brassica* sp. Heterofloral sample: *Cistus* sp. (Cistaceae), *Verbascum* sp. (Scrophulariaceae), *Trifolium* sp. (Leguminosae), *Prunus* sp. (Rosaceae), *Rubus* sp. (Rosaceae), *Asphodelus* sp. (Liliaceae), and *Persea americana* (Lauraceae)	Aqueous extract	DPPH	IC_50_ = 233.3 ± 6.1 μg/mL	[147]
			ABTS	IC_50_ = 56.2 ± 0.8 μg/mL	
Italy	Genus: *Hedera*, *Helianthus*, *Cistus*, *Cornus*, *Brassica*, *Gledistia*, *Hedysarum*, *Trifolium*, *Castanea*, *lamium*, *Magnolia*, *Fraxinus*, *Papaver*, *Crataegus*, *Prunus*, *Rubus*, *and Cordiandrum*	Aqueous/methanol extract	ORAC	839.5 ± 49.5 μmol TE g^−1^ DW	[140]
			ABTS	224.6 ± 18.6 μmol TE g^−1^ DW	
			DPPH	134.7 ± 4.3 μmol TE g^−1^ DW	
Korea	Monofloral samples: *Quercus palustris, Actinidia arguta, Robinia pseudoacacia, and Amygdalus persica.*	Ethanolic extract	DPPH	EC_50_ = 292.0 ± 13.05 μg/mL	[170]
Portugal	*Cistus ladanifer*, *Echium* spp., *Apiaceae*, and *Cistaceae*	Hydroethanolic extract	DPPH	EC_50_ = 2.62 ± 0.09 mg/mL	[146]
			Reducing power Assay	6.51 ± 0.30 mg GAE/mL	
Portugal	*Cistacae Boraginacae*, *Rosaceae*, *Fagaceae*, *Asteraceae*, *Fabaceae*, *Ericaceae*, *Mimosaceae*, and *Myrtaceae*.	Methanolic extract	DPPH	EC_50_ = 3.0 ± 0.7 mg/mL	[155]
			β-carotene bleaching Assays	EC_50_ = 4.6 mg/mL	
Spain	*Cistaceae, Fabaceae, Cistaceae, Ericaceae, Fabaceae, Cistaceae, Ericaceae, and Boraginaceae*	Methanolic extract	DPPH	EC_50_ = 2.98 ± 0.47 mg/mg extract	[164]
			TBARS	EC_50_ = 0.35 ± 0.02 mg/mg extract	
Turkey	ND	Methanolic extract	FRAP	11.77 ± 0.63–105.06 ± 0.59 mmol Trolox/g pollen	[167]
			DPPH	SC_50_ = 0.65–8.20 mg/mL	
			CUPRAC Assay	33.1 ± 0.4–91.8 ± 1.8 mmol Trolox/g pollen	
Turkey	*Centaurea* sp, *Lotus* sp., *Coronilla* sp., *Centaurea* sp., *Scabiosa* sp., *Euphorbia* sp., *Echium* sp., *Coronilla* sp., *Teucrium* sp., *Crepis* sp., and *Castanea sativa*	Ethanolic extract	ABTS	0.373 ± 0.015–5.980 ± 0.100 mg TEAC/g	[160]
			DDPH Assays	1.293 ± 0.031–3.85 ± 0.030 mg TEAC/g	
Turkey	Commercial bee pollen	Extractable fraction	CUPRAC Assay	6.25–64.88 μmol TE/g	[152]
			ABTS	6.20–38.20 μmol TE/g	
			DPPH	0.44–22.45 μmol TE/g	
		Hydrolysable fraction	CUPRAC Assay	69.16–192.96 μmol TE/g	
			ABTS	37.63–80.49 μmol TE/g	
			DPPH	33.21–62.37 μmol TE/g	
		Bio-accessible fraction	CUPRAC Assay	83.24–257.27 μmol TE/g	
			ABTS	48.96–111.40 μmol TE/g	
			DPPH	35.69–83.84 μmol TE/g	
Turkey	ND	Methanolic extract	CUPRAC Assay	0.02 ± 0.02–0.24 ± 0.04 mmol Trolox/g	[166]
			FRAP	8.69 ± 1.64–84.89 ± 10.09μmol FeSO4.7H2O/g	
			DPPH	SC_50_ =0.47 ± 0.51–0.84 ± 0.17 mg/mL	
Poland	*Aesculus hippocastanum*, *Chamerion angustifolium*, *Lamium purpureum*, *Lupinus polyphyllus*, *Malus domestica*, *Phacelia tanacetifolia*, *Pyrus communis*, *Robinia pseudoacacia*, *Sinapis alba*, *Taraxacum officinale*, *Trifolium* sp., and *Zea mays.*	Pepsin-digested extract	DPPH	EC_50_ = 20.912 ± 0.821 μL/mL	[159]
			ABTS	1.752 ± 0.024 mmol Trolox/g	
Malaysia	ND	Ethanolic extract	DPPH	Antioxidant activity = 39%	[163]
Serbia	*Helianthus annuus* L.	Methanolic extract	ABTS	Antioxidant activity = 95.5%	[158]
			FRAP	A700 nm = 0.738	
		Ethanolic extract	ABTS	Antioxidant activity = 75%	
			FRAP	A700 nm = 0.485	
Slovakia	*Helianthus annuus* L.	Ethanolic extract	DPPH	Antioxidant activity = 47.97 ± 0.29–50.46 ± 0.43%	[153]
Slovakia	Monofloral samples: *Brassica napus* L. *var. napus*, *Helianthus annuus* L., *Papaver somniferum* L., *Phacelia tanacetifolia* L., *Robinia pseudoacacia* L., and *Trifolium repens* L.	Methanolic extract	ABTS	0.83 ± 0.10–2.08 ± 0.25 mm/l	[148]
			DPPH	Antioxidant activity = 25.96 ± 1.61–93.69 ± 5.80%	
		Aqueous extract	DPPH	Antioxidant activity = 19.66 ± 1.06–50.29 ± 3.05%	
Slovakia	Monofloral samples: *Brassica napus* subsp. *napus* L, *Papaver somniferum* L., *and Helianthus annuus* L	Ethanolic extract	DPPH	Antioxidant activity = 70.05± 17.17%	[154]
			Reduction power	3575.56 ± 749.04 μg. mL^−1^	
Thailand	Commercial bee pollen	Ethanolic extract	DPPH	40.69 ± 3.01 mg GAE/g extract	[156]
		Aqueous extract	DPPH	21.27 ± 2.63 mg GAE/g extract	
Bosnia and Herzegovina	*Poaceae* spp., Trifolium spp., *Zea mays*, and *Plantago* spp.	Methanolic extract	DPPH	IC_50_ = 1.43 ± 0.00 mg/g	[143]
			FRAP	4.111 ± 0.136 mmol Fe^+2^/g	
			ABTS	Antioxidant activity = 86.13 ± 2.28%	

ND: Not determined.

## Data Availability

Data are available upon request.

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
