# Peer review of "Bee Pollen as Functional Food: Insights into Its Composition and Therapeutic Properties"

_antioxidants, 2023, doi:10.3390/antiox12030557_

Round 1
Reviewer 1 Report
This manuscript provide a comprehensive overview of composition and therapeutic properties of Bee pollen. Thereview was well conceived and prepared. Thus, this work is suitable for publication after some modifications. However, I think this review is more suitable for publication in Foods.
1. Please check the whole manuscript to avoid any errors, for example, “however” should be “However” in Line 94.
2. Some figures could be presented to summarize the therapeutic properties of bee pollen.
3. Some perspectives should be summarized and presented in the review paper.
4. In my opinion, this review is more suitable for publication in Foods.
Author Response
Comments and Suggestions of reviewer 1
This manuscript provides a comprehensive overview of composition and therapeutic properties of Bee pollen. The review was well conceived and prepared. Thus, this work is suitable for publication after some modifications. However, I think this review is more suitable for publication in Foods.
- Please check the whole manuscript to avoid any errors, for example, “however” should be “However” in Line 94.
Answer: Thank you for your remark; the modification is done.
- Some figures could be presented to summarize the therapeutic properties of bee pollen.
Answer: Thank you for your suggestion, the figure has been added, please see page 25.
- Some perspectives should be summarized and presented in the review paper.
Answer: Thank you for your comment, perspectives have been added, please see line 638-653.
- In my opinion, this review is more suitable for publication in Foods.
Answer: Dear reviewer, thank you for your opinion. Despite the food aspect of Bee pollen, the aim of this review is totally within the scope of the present Special Issue "Antioxidant Activity of Honey Bee Products". Our manuscript summarizes the general composition of bee pollen as a well-known hive product and collect some data on polyphenolic compounds used to fight oxidative stress-related diseases, and discusses the implicated mechanisms of action, which will bring added value to this special issue.

Reviewer 2 Report
1) The sentence starting on line 244 is vague and difficult to understand. Please revise to make the point clear.
2) Line 366 has a chemical name typo "furfurfural" should be changed to 'furfural'
3) The organization of Tables 2 & 3 by the country where the pollen sample originated is not useful. Can the authors instead provide information on the family or genus of plant from which the pollen originated? If not, use a recognized climatic region identifier as the category, rather than country.
4) Generally the manuscript needs a thorough editing for English language proficiency as many sentences are fragmentary.
Author Response
Comments and Suggestions of reviewer 2
- The sentence starting on line 244 is vague and difficult to understand. Please revise to make the point clear.
Answer: thank you for your attention; the sentence has been clarified, please see line 250-253.
- Line 366 has a chemical name typo "furfurfural" should be changed to 'furfural'.
Answer: thank you for your remark; the modification is done.
- The organization of Tables 2 & 3 by the country where the pollen sample originated is not useful. Can the authors instead provide information on the family or genus of plant from which the pollen originated? If not, use a recognized climatic region identifier as the category, rather than country.
Answer: Thank you for your comment, the most articles review cited the geographical origin (countries) of the different studied samples to facilitate the samples discrimination. Otherwise, the botanical origin was added in both tables 2 and 3.
- Generally, the manuscript needs a thorough editing for English language proficiency as many sentences are fragmentary.
Answer: Thank you for your attention, the paper has been revised and modifications have been added in the text.

Reviewer 3 Report
The review is well written and organized, there are some language style errors that should be checked (e.g. starting a sentence without capital letter, some compounds with capital letter).
Authors should mention at the end of introduction section which is the novelty of this review comparatively to what was previously published.
Page 3 Lines 84-87: the genus of microorganisms should be in italics.
Author Response
Comments and Suggestions of reviewer 3
The review is well written and organized, there are some language style errors that should be checked (e.g., starting a sentence without capital letter, some compounds with capital letter).
- Authors should mention at the end of introduction section which is the novelty of this review comparatively to what was previously published.
Answer: thank you for your remark; the modification is done, please see: line 44-47.
- Page 3 Lines 84-87: the genus of microorganisms should be in italics
Answer: thank you for your remark; the modification is done.
